# Synthesis of Lithium Phosphorus Oxynitride (LiPON) Thin Films by Li_3_PO_4_ Anodic Evaporation in Nitrogen Plasma of a Low-Pressure Arc Discharge

**DOI:** 10.3390/membranes12010040

**Published:** 2021-12-28

**Authors:** Nikolay Gavrilov, Alexander Kamenetskikh, Petr Tretnikov, Alexey Nikonov, Leonid Sinelnikov, Denis Butakov, Viktor Nikolkin, Andrey Chukin

**Affiliations:** 1Institute of Electrophysics of the Ural Branch of the Russian Academy of Sciences, 620016 Ekaterinburg, Russia; gavrilov@iep.uran.ru (N.G.); tpetr@iep.uran.ru (P.T.); nikonov@iep.uran.ru (A.N.); 2Joint Stock Company “Institute of Nuclear Materials”, 624250 Zarechny, Russia; sinelnikov_lp@irmatom.ru (L.S.); butakov_ds@irmatom.ru (D.B.); Nikolkin.irm@rambler.ru (V.N.); 3Institute of Physics and Technology, Ural Federal University, 620062 Ekaterinburg, Russia; achukin@e1.ru

**Keywords:** LiPON thin films, anodic evaporation, low-pressure arc, plasma diagnostic

## Abstract

Thin amorphous films of LiPON solid electrolyte were prepared by anodic evaporation of lithium orthophosphate Li_3_PO_4_ in an arc discharge with a self-heating hollow cathode at a nitrogen pressure of 1 Pa. Distribution of the arc current between two electrodes having an anode potential provided independent control of the evaporation rate of Li_3_PO_4_ and the density of nitrogen plasma. Stabilization of the evaporation rate was achieved using a crucible with multi-aperture cover having floating potential. The existence of a threshold value of discharge current (40 A) has been established, which, upon reaching ionic conductivity over 10^−8^ S/cm, appears in the films. Probe diagnostics of discharge plasma were carried out. It has been shown that heating the films during deposition by plasma radiation to a temperature of 200 °C is not an impediment to achieving high ionic conductivity of the films. Dense uniform films of LiPON thickness 1 μm with ionic conductivity up to 1 × 10^−6^ S/cm at a deposition rate of 4 nm/min are obtained.

## 1. Introduction

The interest in thin-film solid electrolytes is brought about by their prospects in all-solid-state power supplies for micro- and nano-electronics and microsystem technology [1]. A solid electrolyte based on lithium phosphorus oxynitride (LiPON) has a sufficiently high ionic conductivity (>10^−6^ S/cm), wide electrochemical voltage window (0–5.5 V), high electronic resistance (>10^14^ Ω·cm), and is chemically stable [2,3]. To date, LiPON films have been synthesized using various plasma-assisted deposition methods, including pulsed laser deposition [4], vapor deposition with ion beam assistance [5], electron beam evaporation [6], and plasma chemical vapor deposition [7]. The method of RF-magnetron sputtering is most widely used. It provides the best morphology of LiPON films, high ionic conductivity, and is compatible with integrated circuit manufacturing technologies [2]. The main disadvantage of the method is the low deposition rate of films (usually less than 2 nm/min), which is due to the low specific power of RF discharge on the surface of the sputtered target (2 W/cm^2^) made of lithium orthophosphate (Li_3_PO_4_). When the threshold power is exceeded, the target is destroyed due to the low thermal conductivity of Li_3_PO_4_. Numerous studies aimed at optimizing such parameters of RF magnetron sputtering process as RF power, nitrogen pressure, substrate temperature, bias voltage, annealing conditions of films after deposition, and composition of the sputtered target often give opposite results, which makes it difficult to understand the essence of the phenomena and slows down the development of the method [8].

The electron beam evaporation with plasma assistance provides higher deposition rates. Such a method, where a Li_3_PO_4_ target was vaporized by an electron beam with a power of 300 W, and an inductively coupled RF discharge (ICP) with a power of 250 W in Ar-N_2_ medium was used for plasma activation, as described in [9]. LiPON films with Li-ion conductivity of 10^−7^–10^−8^ S/cm were obtained by this method at a deposition rate of 8.33 nm/min. The authors concluded that an increase in RF discharge power, leading to an increase in the concentration and kinetic energy of nitrogen particles in plasma, was a key factor in increasing the ionic conductivity of the deposited films. However, with a significant increase in power, conditions for the crystallization of thin films are created, which reduces the ionic conductivity of LiPON film. The authors consider the main direction of improving the method as increasing the density of nitrogen plasma while maintaining low ion energy.

In another embodiment of the electron beam evaporation method, Li_3_PO_4_ vapors were transported in the direction of the substrate using a supersonic gas (He+N_2_) jet. Ar plasma was created near the substrate using a discharge with a hollow cathode, which provided nitrogen ionization and its reactive injection into Li_3_PO_4_ film [6]. The deposition rate of the LiPON film reached 178 nm/min, the maximum value of the ionic conductivity of the films was 5.2 × 10^−7^ S/cm. The conductivity of the films increased with an increase in the nitrogen flux. However, its growth was limited by a decrease in the ionic conductivity of the films due to an increase in lithium losses and partial crystallization of the films. Despite the achieved high deposition rate of LiPON films, this method has not been widely used due to the complexity of its technical implementation.

As a new method of thin LiPON films deposition the anode evaporation of lithium orthophosphate (Li_3_PO_4_) in nitrogen plasma of a low-pressure arc with a self-heated hollow cathode may be used. Adjustable distribution of the arc current between two electrodes having anode potential provides both controlled heating of the anode-crucible and generation of nitrogen plasma with the required density. The method ensured a high rate (up to 5 μm/h) of low temperature (500–600 °C) deposition of α-Al_2_O_3_ films by reactive evaporation of Al in a low-pressure Ar/O_2_ media [10]. However, the use of this method for evaporation of Li_3_PO_4_ and high-rate LiPON thin-film deposition is difficult due to the impossibility of direct heating of a non-electroconductive Li_3_PO_4_ by electron flow from the plasma, as well as significant heating of the deposited film with radiation from the discharge plasma, which can lead to LiPON crystallization and a decrease in ionic conductivity of a film [11].

The present paper defines discharge parameters and gas conditions under which synthesis of thin films (~1 μm) having ionic conductivity (~1 × 10^−6^ S/cm) is ensured. The method of indirect heating of Li_3_PO_4_ powder is described, providing its stable evaporation with controlled rate. The temperature of the substrate during the film deposition was measured and it was shown by XRD analysis that the films are X-ray amorphous. Probe diagnostic of plasma was used to measure the electron temperature and values of potential drop in layers of space charge near the crucible and substrate. The effect of nitrogen pressure and discharge current on the degree of nitrogen dissociation in plasma was evaluated by optical emission spectroscopy. The morphology of the film’s surface was observed by SEM.

## 2. Materials and Methods

The deposition of thin LiPON films was carried out in an electrode system (Figure 1), including a self-heating hollow TiN cathode 1 [12], through which Ar or N_2_ (50 sccm) was pumped, a water-cooled anode 2 placed inside a cylindrical shielding electrode 3, through the cavity of which N_2_ (100–150 sccm) was fed into the discharge gap, and a graphite or tantalum crucible 4. The design of the hollow cathode ensured its long-term operation at discharge currents up to 100 A [13]. The shield electrode restricts the cross-section of the discharge gap in the anode region, which contributes to an increase in the molecular gas dissociation degree [14]. The discharge current was distributed between the anode 2 and the crucible 4, providing regulated heating and evaporation of Li_3_PO_4_ and generation of dense N_2_ plasma in the volume. The vacuum chamber was pumped out by a turbomolecular pump at a speed of 500 L/min, the partial N_2_ pressure in the deposition process was 1 Pa.

Li_3_PO_4_ powder (purity 99.5%) was used in experiments; an amount of 0.3 g of powder was loaded into a crucible with an inner diameter of 10 mm. The evaporation rate of the powder was controlled by the amplitude of the line of the excited LiI atom (670.8 nm) in the discharge plasma emission spectrum using an HR2000 spectrometer (Ocean Optics, Inc., Winter Park, CO, USA) and was regulated by changing the current in the crucible circuit. The films were deposited on polished 12Cr18Ni10Ti stainless steel substrates mounted on a multi-position sample holder 5, the temperature of which was controlled using a thermocouple and Impac IP 140 optical pyrometer (LumaSense Technologies, Santa Clara, CA, USA). The sample holder was under a floating or low (−10 V) negative bias potential. An ohmic heater 6 was installed on the back of the holder for the thermal annealing of deposited films. The temperature of the substrates during the deposition of films was determined mainly by the thermal effect of the discharge plasma and reached 200 °C at a discharge current of 40 A. The distance between the sample holder and the crucible was ~100 mm. After the films were deposited and the samples cooled, they were moved to a magnetron sputtering chamber, in which electrically conductive stainless steel contact layer with a thickness of 0.2 μm was deposited to the film surface.

The instability of evaporation of Li_3_PO_4_ powder from an open crucible is caused by changes in discharge parameters such as the electron temperature and the anode potential drop when the composition of the vapor-gas medium changes. Since these parameters affect the power of the crucible heating, the occurrence of local fluctuations in the vapor density or current density distribution over the outer surface of the crucible can lead to short-term pressure surges, and with low heating power, changes in vapor pressure can occur cyclically with a frequency of fractions of Hz. Along with short-term instabilities of the evaporation rate, its value also decreases with time, since as the mass of the melt in the crucible decreases, the heat exchange surface area at the wall surface decreases. Another feature of the method of anodic evaporation in an arc discharge is the dependence of the degree of ionization of the material vapors in the flow of fast electrons on the current value in the crucible circuit. It can have a significant effect on the structure of the deposited film and the value of the ionic conductivity of LiPON films.

To stabilize the evaporation rate of Li_3_PO_4_ and reduce the dependence of the degree of ionization of the vapor flow on the heating current of the crucible, some steps were adopted to separate the vapor flow and the electron flow in near-crucible area. For this purpose, the crucible is closed with an electrically insulated cover with small holes (diameter of 2 mm). The floating potential of the cover does not allow the discharge current to be directly closed to the lid. The smallness of the holes does not allow the discharge to burn in the steam flow through the holes to the inner surface of the crucible walls. Local gas supply to the area of the lateral surface of the crucible helps to hold the discharge in this area. The long-term instability of the evaporation rate was eliminated by introducing a negative feedback between the heating current of the crucible and the amplitude of the line of the excited LiI atom in the plasma optical emission spectrum. The part of the spectrum near the lithium atom line used to control the evaporation rate of Li_3_PO_4_ is shown in Figure 2.

Along with controlling the evaporation rate of Li_3_PO_4_, it is important to provide conditions for the injection of nitrogen into the deposited film. A key factor in this process is the high concentration of atomic nitrogen. To increase the degree of N_2_ dissociation, geometric compression of the arc column in the near-anode region is used, through which N_2_ is fed into the discharge gap. In such an electrode system, potential jumps occur at the entrance to the constriction and near the anode, in which electrons acquire energy that ensures effective dissociation of gas molecules by an electron impact [14].

The influence of the parameters and conditions of the discharge burning on the degree of N_2_ dissociation was evaluated by optical actinometry [15]. During measurements, 5% Ar was added to N_2_, and the ratio of the intensities of atomic nitrogen lines N and Ar was used for calculations. The choice of spectrum lines should meet a number of conditions, in particular, the excitation of particles must occur by an electron impact from the ground state, the cross-sections of the particle excitation processes must have the same shape and close values of the excitation threshold, and the absorption of radiation in plasma at wavelengths corresponding to the lines of the particles used should be minimal. In [16], it was shown that the NI line with a wavelength of 746.8 nm and the ArI line (750.4 nm) met these conditions.

Probe diagnostics of plasma were used to determine the electron temperature and plasma potential. The measurement results were used to calculate the values of near-electrode potential drops in the positive space charge layer near the crucible and in the negative space charge layer near the floating sample holder. The voltage drop in the layer near the crucible, along with the current in the crucible circuit, affects the heating power of the crucible, and the voltage drop in the layer near the sample holder affects the energy of ions entering the surface of the film. The ion energy may be increased by applying a bias voltage to the substrate. For diagnostics, a Langmuir collecting probe made of tungsten wire with a diameter of 0.6 mm and length of 5 mm was used.

The ionic conductivity of LiPON films in symmetrical capacitor cells “steel/LiPON/steel”, obtained after deposition of stainless steel 12Cr18Ni10Ti contact by the magnetron sputtering method, was measured by electrochemical impedance spectroscopy [17]. For impedance measurements, R-40X (Chernogolovka, Russia) was used; the signal amplitude was 0.15 V; the signal frequency was varied within the range 0.5–5 × 10^5^ Hz. The obtained impedance spectra were analyzed using ZView software.

X-ray phase analysis of Li_3_PO_4_ powder and LiPON films was carried out on a D8 DISCOVER diffractometer in copper radiation (Cu Kα1,2 λ = 1.542 Å) with a graphite monochromator on a diffracted beam. The spectra were processed using the TOPAS 3 software with the Rietveld algorithm for clarifying structural parameters. The thickness of LiPON films was estimated by the gravimetric method, as well as by ball-cratering abrasion test. The morphology of the film surface was studied using an Olympus BX41M-LED optical microscope and a scanning electron microscope (JSM-6390LV, Jeol).

## 3. Results

The discharge current to the crucible necessary to achieve the required evaporation rate of Li_3_PO_4_ after it is heated above the melting point (1220 °C) was determined at N_2_ flow rate of 90 sccm through a hollow cathode, ensuring stable discharge operation. The film deposition rate of 0.25 μm/h was obtained at a current of 10 A in a graphite crucible circuit with an output aperture area of 0.8 cm^2^. Then a series of experiments were carried out at different values of the current in the anode circuit and N_2_ flow rate. Films with an ionic conductivity of ~10^−6^ S/cm were obtained at an anode current of 40 A, a N_2_ flow of 250 sccm, and a negative bias voltage of 10 V applied to the substrate. The value of the threshold current to the anode, at which the ionic conductivity of the films over 10^−8^ S/cm occurs, correlates with the rate of evaporation of Li_3_PO_4_. The temperature of the metal substrates under these conditions reached 216 °C, and heating was provided mainly by radiation from the discharge plasma. The average ion current density per samples from plasma (~3 mA/cm^2^) was determined by the ion current in the sample holder circuit, measured with an increase in the negative bias voltage to the values of ~100 V, at which the ion current saturated.

The impedance spectrum for the LiPON film obtained in the mode of stable evaporation of Li_3_PO_4_ from a non-equipotential crucible at the above parameters of the deposition process is shown in Figure 3. The inset in Figure 3 shows the equivalent circuit for the treatment of the experimental data. The resistance *R_S_* is due to the contact resistance in the measuring circuit and characterizes the shift of the impedance spectrum from the origin along the *Z_Re_* axis. The high-frequency part of the spectrum described by a parallel-connected resistance *R_1_* and a constant phase element *Q_1_* corresponds to the ionic transfer in the LiPON film [18]. The low-frequency part of the impedance spectrum described by the *Q_2_* element corresponds to the polarization of the electrodes [19]. The ionic conductivity (σ) of LiPON films was calculated using the relation σ = (1/*R_1_*) × (δ/*A*), where δ is the LiPON film thickness (~1 μm) and *A* is the area of the deposited metal electrode (~1.3 cm^2^).

The microstructure of LiPON films obtained in the initial experiments was characterized by a combination of two different structures: large formations with a size of 5–10 µm and spiral formations located in the space between them, similar to those described earlier in [20]. By optimizing the evaporation conditions, increasing N_2_ flow rate and applying a bias voltage (−10 V), it was possible to form denser and more uniform films with a smaller size of large-scale elements of the film structure (Figure 4), which, nevertheless, significantly exceeds the characteristic size of elements in films obtained by magnetron sputtering [8].

Diffractograms of LiPON films deposited at 200 °C for 2 h, not annealed (1) and annealed in vacuum at temperatures 200 (2), 250 (3) and 300 °C (4) are shown in Figure 5. The films were stored in N_2_ and came into contact with the atmosphere only during measurements (2 h). All films are amorphous; however, lines of lithium monohydrate LiOH·H_2_O are observed in the angles between 30° and 38°; the appearance of a weak Li_3_PO_4_ (22.4°) line is characteristic of annealed films. Thus, it can be concluded that deposition of LiPON at an elevated temperature does not lead to crystallization of the films, or they are in an X-ray amorphous state. Even short-term contact with the atmosphere leads to the appearance of lithium hydroxide. Two-hour annealing at a temperature of 300 °C leads to the appearance of traces of the crystalline phase Li_3_PO_4_ (plot 4), which are absent for all other films. Tests have shown that annealed films retain their properties longer during storage and, as a rule, have less LiPON–metal interface resistance.

The dependencies of the plasma electron temperature on Li vapor density for two crucible heating options obtained by the Langmuir double probe method are shown in Figure 6. The temperature of electrons during evaporation from an equipotential crucible is almost constant (2.2 eV). When using a crucible with an insulated multi-aperture lid, the temperature decreases slightly with increasing vapor pressure.

The plasma potential was determined from the inflection of the current-voltage characteristic of a single Langmuir probe at the sign change point of the second derivative [21]. The dependencies of the potential difference between the crucible and the plasma (Δφ_a_) on Li vapor pressure for two different crucibles are shown in Figure 7. The electrons are accelerated by the potential drop in the positive space charge layer to approximately a constant value of 26.5 eV in the case of a probe with an insulated lid. The energy of the directed movement of electrons in the flow to the probe with a conductive cover increases slightly with the vapor pressure. For a more accurate assessment of the electron energy on the crucible surface, it is necessary to take into account their thermal energy and the work of the electron output from the crucible material [22]. On average, the heating power at a crucible current of 10 A was about 350 W. The measured potential difference in the negative space charge layer between the plasma and the floating substrate (Δφ_f_), in which the ions bombarding the surface of the growing film acquire energy, is ranged from 11 to 18 V (Figure 7). When a negative bias voltage U_b_ is applied to the samples, the ion energy increases additionally by the amount of eU_b_.

During actinometric measurements, a difference spectrum obtained by subtracting the measured values of the discharge plasma spectrum in pure N_2_ from the data for the discharge plasma spectrum in N_2_ + 5% Ar was used to more accurately determine the amplitude of the ArI line (750.4 nm) against the background of intensity of the first positive N_2_ system (B^3^Π_g_—A^3^Σ_u_^+^). Calculations of the degree of nitrogen dissociation [N]/(2[N_2_]), carried out using coefficients from [16], gave the ratio values of ~27–35%. The dependencies of [N]/(2[N_2_]) on the magnitude of the discharge current and N_2_ pressure are shown in Figure 8.

## 4. Discussion

Anodic evaporation of Li_3_PO_4_ makes it possible to use higher power densities compared to magnetron sputtering and generate denser vapor flows. At the same time, in order to ensure high growth rates of LiPON film with high ionic conductivity, it is necessary, first, to optimize such parameters, of the flow of nitrogen particles from the plasma to the surface of the growing coating, as flow density, and degree of its ionization, and dissociation. The task of obtaining the optimal elemental composition of the film will be solved at the next stages of this study. Since the intensive ionic assistance of the deposition process contributes to the crystallization of the amorphous film at low temperatures [23], it is desirable to reduce the degree of ionization of the vapor-gas medium, which is achieved by using a non-equipotential crucible. To increase the degree of N_2_ dissociation, the method of contraction of the anode part of the arc, which has been tested earlier, was used [24].

Another difference between the method of anodic evaporation and magnetron sputtering is the increased (about 200 °C) temperature of the films. Studies of the effect of post-annealing of magnetron sputtering deposited LiPON films on the value of their ionic conductivity have been carried out in a number of studies. Thus, heating LiPON film at a rate of 30 °C/min to 260 °C with exposure at this temperature for 2 min and subsequent natural cooling led to an increase in the ionic conductivity of the films from 3.0 × 10^−6^ S/cm to 7.2 × 10^−6^ S/cm [25]. Annealing of films at a temperature of 200 °C was accompanied by an increase in ionic conductivity from 2.3 × 10^−6^ S/cm at 20 °C to 3.2 × 10^−6^ S/cm [26]. The study of the effect of post-annealing of films in the temperature range from 100 to 400 °C showed an increase in ionic conductivity from 1.26 × 10^−6^ to 3.31 × 10^−6^ after 3 h of annealing at a temperature of 200 °C [27]. Annealing at higher temperatures worsened the properties of the films. Post-annealing at 200 °C led not only to a decrease in impedance but also improved the cyclic stability of the films [28]. Thus, it can be assumed that the increased temperature of the films during their deposition is not a factor in preventing the achievement of their high ionic conductivity.

In the study devoted to the measurement of plasma parameters when applying LiPON films by magnetron sputtering, it was shown that high-quality films with high ionic conductivity, compact and smooth structure were obtained at low gas pressures and moderate discharge power [29]. Under these conditions, the discharge is characterized by low plasma density, high electron temperature, high plasma potential, and a long path length of the atomized particles. Measurements of the N/N_2_ amplitudes ratio in the plasma mass spectrum by appearance mass spectrometry showed that at low gas pressure and low discharge power, the ratio N/N_2_ reached 20, which indicates a high degree of N_2_ dissociation in the discharge. The electron temperature in the discharge was in the range of 1–1.5 eV, the plasma potential was about 9 V. With a decrease in pressure from 50 to 5 mTorr, the ionic conductivity of the films increased from 0.47 to 2.16 μS/cm, and the size of the structural elements on the surface of the film was decreased by an order of magnitude, from several μm to fractions of μm.

It is incorrect to carry out direct comparisons of plasma parameters of arc and magnetron discharges and draw conclusions on this basis about ways of optimizing the mode of arc deposition of LiPON films, primarily due to the significant difference in the energy of vaporized neutral particles, which is determined by the melt temperature (~0.1 eV), and the atomized particles, which are in the range 1–10 eV [30]. However, it is obvious that it is necessary to achieve a high degree of N_2_ dissociation, which ensures the required level of its content in the film at low gas pressure in the discharge. In addition, it is necessary to optimize the ratio of the densities of evaporated particles and nitrogen flows coming to the surface of the growing film.

The obtained values of the degree of nitrogen dissociation of 27–35% significantly exceed the values measured in inductively coupled plasma (ICP) discharges. Thus, the degree of N_2_ dissociation in Ar/N_2_ plasma of ICP discharge, calculated from the ratio of N (746.8 nm) and Ar (750.4 nm) lines, was about 10% [16]. In [31], actinometric measurements in N_2_-Ar plasma of ICP plasma source were carried out using the ratio of N (746.68 nm) and Ar (750.4 nm) lines. The maximum value of the degree of dissociation of N_2_ (2%) was obtained at a nitrogen pressure of 2 mTorr and decreased to 1% at a pressure of 20 mTorr. A high degree of O_2_ dissociation (up to 0.4) in dense oncoming flows of gas particles and fast electrons existing in an arc with a near-anode narrowing of the discharge gap was previously measured by the catalytic probe method [14].

The importance of measuring the electron temperature in the function of metal vapor pressure is due to the fact that a change in the ratio of gas and metal components in the plasma can lead to a significant change in plasma parameters. The reason is the lower ionization potential of metal atoms (5.2 V-Li) compared to gas (15 V-N). For example, when an electron beam evaporated Cu in a low-pressure gas (He), an increase in the proportion of Cu vapors over 10% led to a sharp drop in the electron temperature of the plasma (from 2.5 to 0.4 eV) and an increase in plasma concentration [32]. It is also known that the method of anodic evaporation in a vacuum arc is capable of providing a high degree of ionization of metal vapors [33]. The high degree of vapor ionization during anodic evaporation of Al (up to 80%), achieved by magnetic compression of the arc column on the melt surface, ensured a decrease in the crystallization temperature of α-Al_2_O_3_ film [34]. It was not possible to carry out direct measurements of the LiII line in the optical emission spectra of plasma due to its low intensity. However, a slight change in the electron temperature over a wide range of changes in the evaporation rate of Li_3_PO_4_ indirectly indicates a low degree of ionization of lithium vapor under experimental conditions, which minimizes the influence of this factor on the phase composition of the LiPON film.

The highest value of the ion conductivity of LiPON film (1 × 10^−6^ S/cm) was obtained at an average ion current density on the coating surface of 3 mA/cm^2^ (film deposition rate of 4 nm/min), which is significantly higher compared to the ion current density of 0.54 mA/cm^2^ during electron beam evaporation in a supersonic flow of N_2_ + Ne [34] and the film deposition rate up to 178 nm/min. At the same time, the densities of nitrogen fluxes to the film surface differ slightly (2.1 × 10^18^ [20] and 4 × 10^18^ molecules/(cm^2^·s) in and the present experiments, respectively). Probably, the increased current density of the ion assistance compensates for the significant difference in the energy of nitrogen atoms entering the film surface. However, an increase in the bias voltage above 10 eV, as in [20], led to a decrease in the ionic conductivity of the films. The main characteristics of electron-beam and anodic arc methods are given in the Table 1.

Thus, the ionic conductivity of LiPON films deposited by arc anodic evaporation (1 × 10^−6^ S/cm) exceeded the values for films deposited by electron beam evaporation (5.2 × 10^−7^ S/cm [20] and 6.0 × 10^−7^ S/cm [9]). However, the deposition rate for anodic arc deposition is relatively low. Optimization of conditions for deposition of LiPON films at high rates will be the subject of further research.

## 5. Conclusions

By the method of anodic evaporation of Li_3_PO_4_ in a low-pressure arc discharge (N_2_, 1 Pa) and vapor condensation under conditions of intense exposure to nitrogen ions (3 mA/cm^2^) and a high degree of N_2_ dissociation in plasma (up to 35%), LiPON solid electrolyte films with a thickness of ~1 μm at a rate of 3–5 nm/min were obtained. The peculiarities of the method are the difficulty of stabilizing the evaporation rate of non-electrically conductive Li_3_PO_4_, the presence of a threshold value of the discharge current, which upon reaching the films acquires ionic conductivity, and the increased temperature of the films caused by intense radiation from the plasma during the deposition process. To maintain a constant rate of Li_3_PO_4_ evaporation, a non-equipotential multi-aperture crucible was used, the heating current of which was corrected using feedback on the amplitude of the lithium atom line in the plasma emission spectrum. The threshold current to the anode correlates with the rate of Li_3_PO_4_ evaporation. LiPON films obtained at a temperature of ~200 °C are X-ray amorphous, the ionic conductivity of the films, measured by impedance spectroscopy, was up to ~1 × 10^−6^ S/cm. The results obtained confirm the efficiency of using the anodic arc evaporation of Li_3_PO_4_ in nitrogen plasma for the deposition of thin LiPON films.

## Figures and Tables

**Figure 1 membranes-12-00040-f001:**
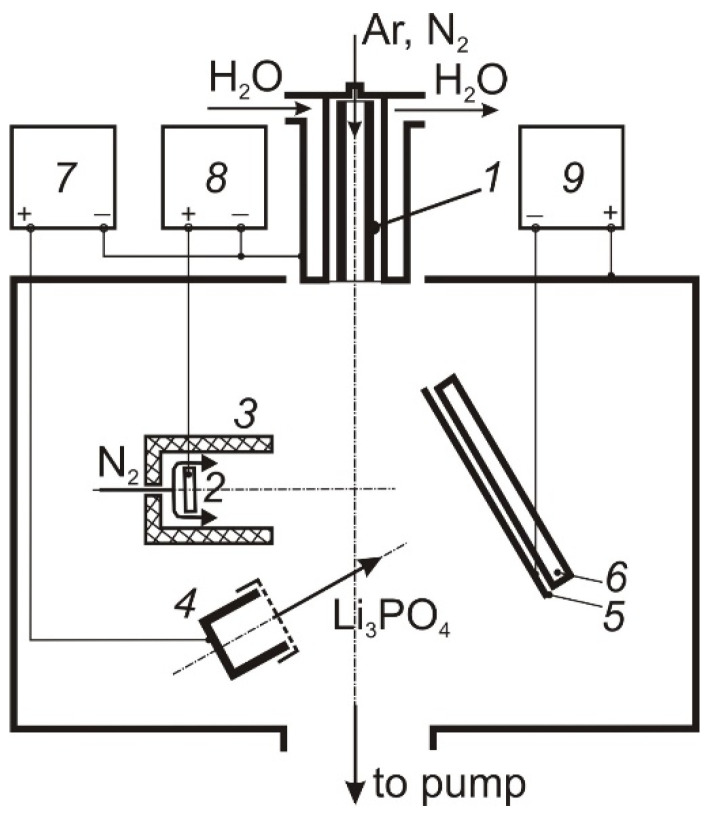
Deposition facility. 1 self-heated hollow cathode; 2 anode; 3 shield; 4 crucible; 5 samples holder; 6 heater; 7, 8, 9 power supply.

**Figure 2 membranes-12-00040-f002:**
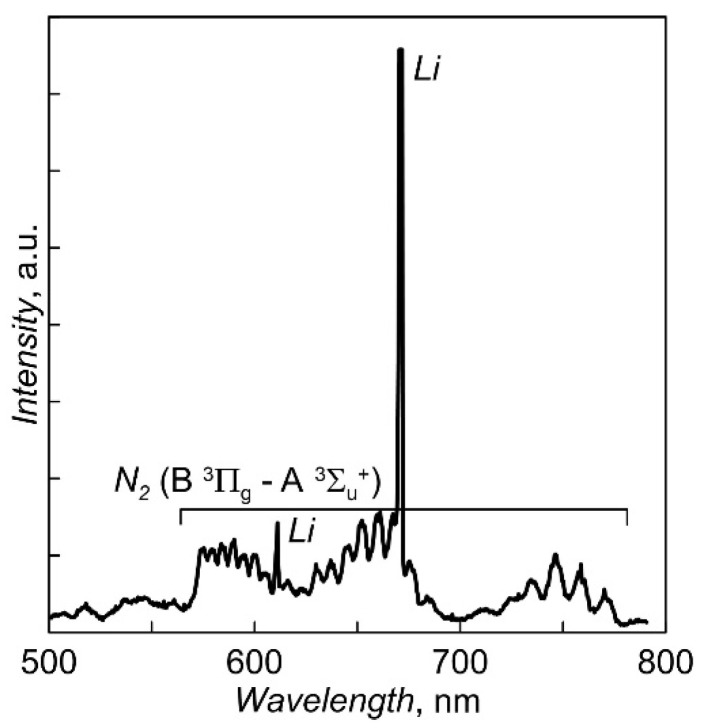
The fragment of optical emission spectrum of discharge plasma under Li_3_PO_4_ evaporation.

**Figure 3 membranes-12-00040-f003:**
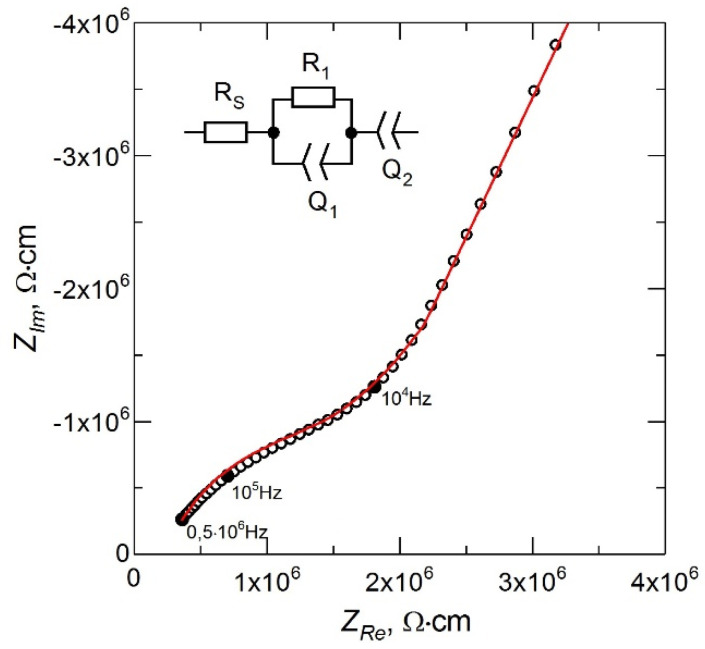
Nyquist plot of LiPON thin film.

**Figure 4 membranes-12-00040-f004:**
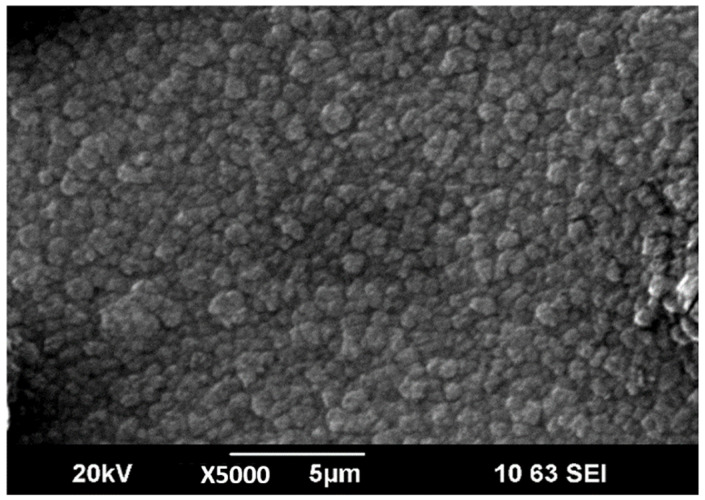
SEM image of surface of LiPON thin-film.

**Figure 5 membranes-12-00040-f005:**
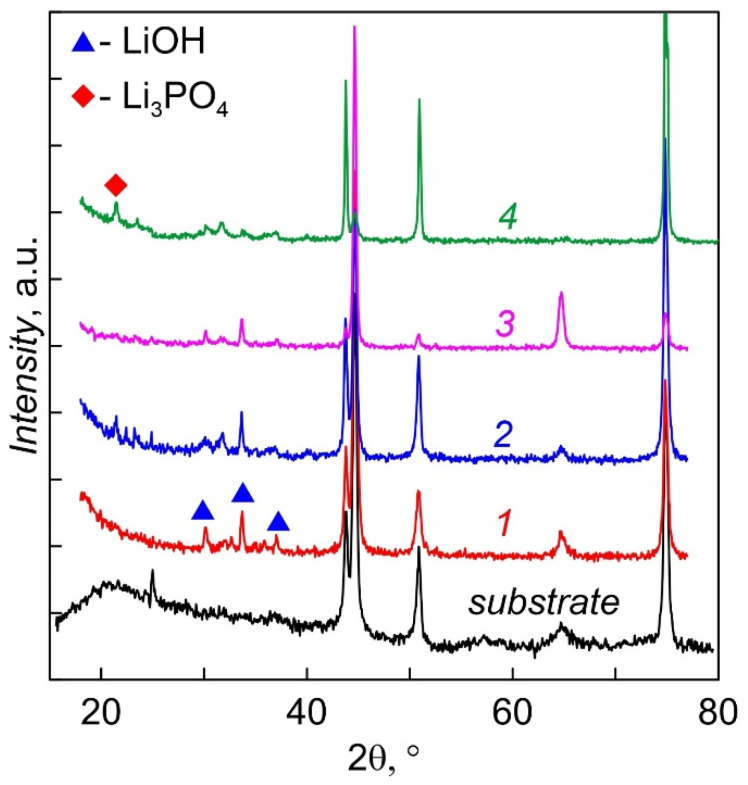
XRD patterns of LiPON films. 1 as-deposited. 2, 3, 4, post-annealed under temperature: 2— 200, 3—250, 4—300 °C.

**Figure 6 membranes-12-00040-f006:**
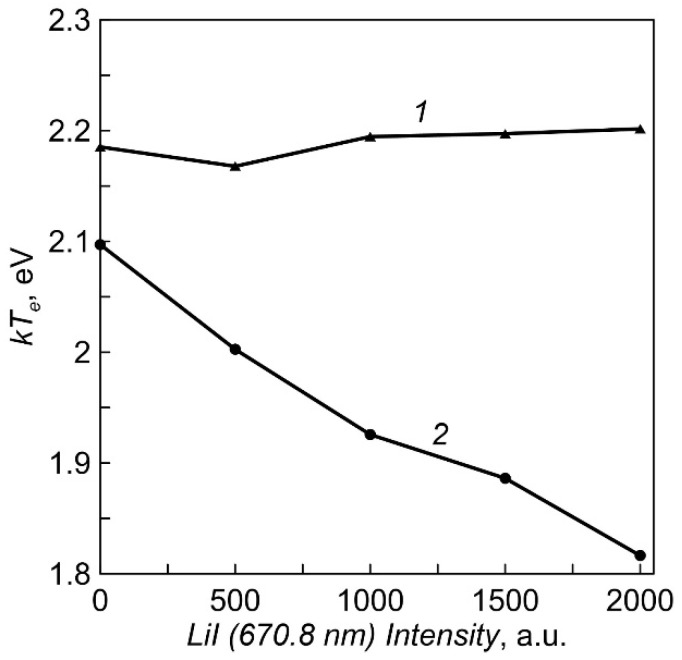
The dependencies of the plasma electron temperature on intensity of LiI line (Li vapor density). *1*—equipotential crucible, *2*—crucible with an insulated multi-aperture lid.

**Figure 7 membranes-12-00040-f007:**
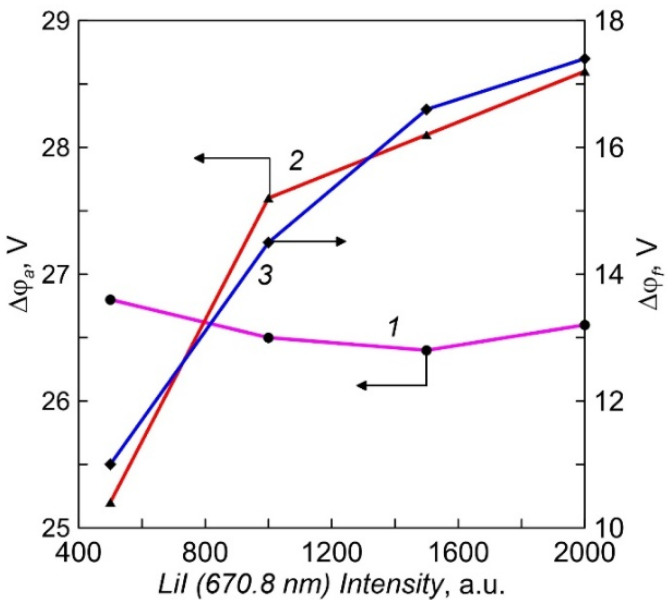
The dependencies of potential difference between the crucible and the plasma (Δφ_a_) and the plasma and the floating substrate(Δφ_f_) on intensity of LiI line (Li vapor density). *1*, *3*—crucible with an insulated multi-aperture lid; *2*—equipotential crucible.

**Figure 8 membranes-12-00040-f008:**
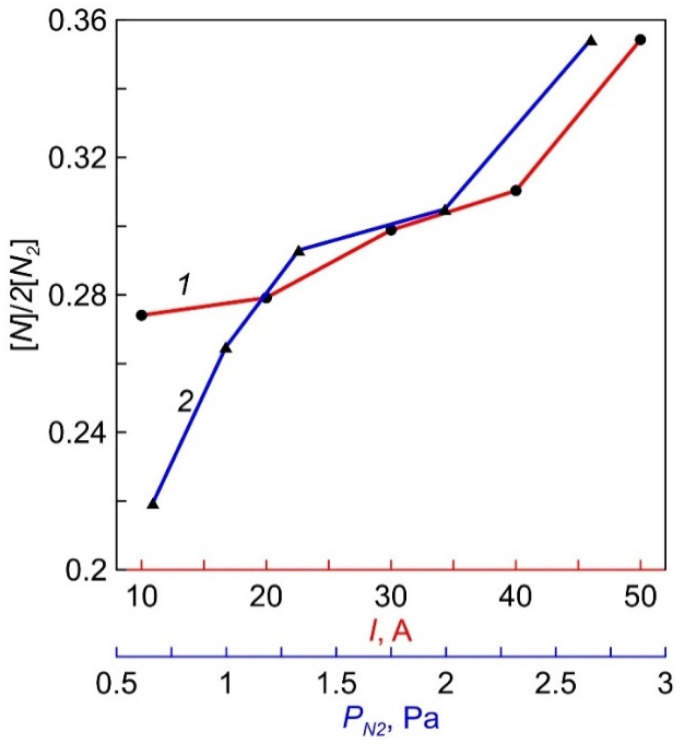
The dependencies of N_2_ dissociation degree on discharge current (1) and N_2_ pressure (2).

**Table 1 membranes-12-00040-t001:** The comparative characteristics of deposition conditions.

Parameter	[9]	[20]	Present Paper
Evaporator type	Electron beam 300 W	Electron beam 160 W	Anodic arc 5 A
Plasma generator	ICP 250–450 W	Arc 60 A	Arc 40 A
Max ionic conductivity, S/cm	6 × 10^−7^	5.2 × 10^−7^	1 × 10^−6^
Max deposition rate, nm/min	8.3	178	5
Substrate temperature, °C	RT	180 ± 20 °C	200 ± 20 °C
Gas pressure, Pa	1 × 10^−2^ N_2_/Ar = 3/8	12–13 N_2_ + He + Ar	1 N_2_
Bias voltage, V	floating	−20	−10

## Data Availability

Not applicable.

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
