# Peer review of "Synthesis of Lithium Phosphorus Oxynitride (LiPON) Thin Films by Li3PO4 Anodic Evaporation in Nitrogen Plasma of a Low-Pressure Arc Discharge"

_membranes, 2021, doi:10.3390/membranes12010040_

Round 1

Reviewer 1 Report

The manuscript entitled “Synthesis of lithium phosphorus oxynitride (LiPON) thin films by Li3PO4 anodic evaporation in nitrogen plasma of a low-pressure arc discharge” by Gavrilov et al. is an interesting study on anodic evaporation for LiPON film fabrication.  This paper is acceptable after minor revision.

I have the following concerns.

  • In the abstract, there is a description mentioning the ionic conductivity over 10−8 S cm−1. The corresponding part seems to be missing in the main text.
  • The frequency information should be inserted in Fig. 3.
  • I recommend fitting the Nyquist plot in Fig. 3 to more rigorously analyze the data.
  • The caption for Fig. 5 does not explain 2, 3, and 4.
  • The caption for Fig. 8 does not explain 2.

Minor concerns

  • In the abstract, S/cm1 should be S/cm.
  • I recommend noting “SEM image” in the caption of Fig. 4.

Author Response

We thank Reviewer for the careful reading of our manuscript and helpful comments that improved presentation and readability of our work. Please find below line-by-line answers to the comments. All changes made in our manuscript are shown in yellow. We hope that revised version of manuscript now is suitable for acceptance  in  Membranes. Please see our answers below 

Answers

1) We have corrected errors in the text pointed out by the reviewer

2) The frequency information inserted in Fig.3

3) We have inserted the fitting in Fig. 3 corresponding to the equivalent electrical circuit and obtained using ZView software. The resistance RS is due to the contact resistance in the measuring circuit and characterizes the shift of the impedance spectrum from the origin along the ZRe axis. The high-frequency part of the spectrum described by a parallel-connected resistance R1 and a constant phase element Q1 corresponds to the ionic transfer in the LiPON film. The low-frequency part of the impedance spectrum described by the Q2 element corresponds to the polarization of the electrodes. The ionic conductivity (σ) of LiPON films was calculated using the relation σ=(1/R1)×(δ/A), where δ is the LiPON film thickness (~1 μm) and A is the area of the depos-ited metal electrode (~1.3 cm2).

4) We have inserted a description in the caption of Fig. 5 and improved text of the paper.

5) We have inserted the description of plot 2 (Fig.8).

6) We noted “SEM image” in the caption of Fig. 4.

 Sincerely yours,

Dr. N.V. Gavrilov on behalf of authors

Reviewer 2 Report

Dear, author.

I read your manuscript gratefully.

Here is my comments.

  1. Authors are compared arc deposition and E-beam deposition at the introduction section. Authors should organize the contents described above in a table
  2. Control groups are unclear. Authors should compared control groups and experimental group clearly. (Fig3. Nyquist plot, Fig4. SEM images)
  3. Authors need to make clearly how to obtain ion current density by using mathematical tactics. (Page 5. Line207)
  4. The magnification looks not enough to be recognize the size of the structure elements. How authors can recognize average particle sizes? (Page 5-6, line 208-214)

Author Response

We thank Reviewer for the careful reading of our manuscript and helpful comments that improved presentation and readability of our work. Please find below line-by-line answers to the comments. All changes made in our manuscript are shown in yellow. We hope that revised version of manuscript now is suitable for acceptance  in  Membranes. Please see our answers below 

Answers

1) The authors agree with the expert's proposal, which will undoubtedly improve the perception of the content of the article, however, it is impossible to carry out a detailed comparison of the electron-beam and arc methods of LiPON deposition in the “Introduction”, since the arc method is used for the deposition of LIPON for the first time. It would be incorrect to make a comparison using the results obtained by the authors earlier for the arc deposition of Al2O3, since this is a completely different material, which required completely different deposition conditions, as mentioned in the “Introduction”. We believe that it is most logical to compare the methods after presenting the results obtained, therefore, we propose to place the table in the "Discussion" section.

2) It is difficult to describe the carried out multivariate experiments in terms of control and experimental groups. At the first stage, the substrate material is selected and several samples are prepared, on each of which films are sequentially deposited at different crucible heating currents. As a result, the dependence of the film deposition rate on the crucible current is obtained. The optimum crucible current is set, and at this constant crucible current, a series of depositions is carried out at different arc currents. The ionic conductivity and film microstructure of this films are measured and the optimal value of the arc current is determined. Then, with a constant crucible current and arc current, the nitrogen flow is varied for a series of samples, the films are analized, and the optimal flow rate is determined. A series of depositions are then carried out for this combination of crucible and arc currents and gas flow at different bias voltages. Measurements are taken. It is clear that the result obtained by this way is not the best, since the variation of each subsequent parameter affects the optimal value of the previous parameter. So, a more detailed selection of parameters is carried out, but already in a narrower range of their variation.

3) The average ion current density per samples from plasma (~3 mA/cm2) was determined by the ion current in the sample holder circuit, measured with an increase in the negative bias voltage to the values of ~ 100 V, at which the ion current saturated.

4) The microstructure of LiPON films obtained in the initial experiments was characterized by a combination of two different structures: large formations with a size of 5-10 µm and spiral formations located in the space between them, similar to those described earlier in [Kim, Y.G.; Wadley, H.N.G. The influence of the nitrogen-ion flux on structure and ionic conductivity of vapor deposited lithium phosphorus oxynitride films. J. Power Sources 2011 196 1371-1377]. By optimizing the evaporation conditions, increasing N2 flow rate and applying a bias voltage (-10 V), it was possible to form denser and more uniform films with a smaller size of large-scale elements of the film structure (Figure 4), which, nevertheless, significantly exceeds the characteristic size of elements in films obtained by magnetron sputtering [Ko, J.; Yoon, Y.S. Infuence of process conditions on structural and electrochemical properties of lithium phosphorus oxynitride thin films. Ceram. Internat. 2020 46 20623-20632].

 Sincerely yours,

Dr. N.V. Gavrilov on behalf of authors

Reviewer 3 Report

The manuscript focused on the elaboration of lithium phosphorus oxynitride thin amorphous films by anodic evaporation of 11 lithium orthophosphate Li3PO4 in an arc discharge with a self-heating hollow cathode at a nitrogen 12 pressure of 1 Pa. The relevance of this work is related to the use of lithium phosphorus oxynitride (LiPON) as an amorphous solid lithium-ion conductor in solid state thin film batteries (TFBs).

The introduction presents important contribution in the field of LiPON thin films elaboration and characterization, important experimental details have been mentioned. The authors focused on the elaboration method and made a clear and deep presentation of the experimental set-up. In my opinion this will strongly contribute to the scientific value of this work. Furthermore, the characterization section includes investigation of the films from structural, morphological and applicative point of view. The results are well correlated, discussed but the conclusions should be improved.

I suggest to the authors to include supplementary explanation in the figure captions especially to figure 5 caption for 1 to 4 diffractograms. Also, the diffractograms number should be mentioned in the text for a better understanding of the explanation.

Eventually, taking into consideration the presented aspects, I recommend the present version of the article for the publication after a minor revision.

Author Response

 We thank Reviewer for the careful reading of our manuscript and helpful comments that improved presentation and readability of our work. Please find below line-by-line answers to the comments. All changes made in our manuscript are shown in yellow. We hope that revised version of manuscript now is suitable for acceptance  in  Membranes. Please see our answers below. 

Answers

1) We improved conclusions.

2) We have inserted the description of all plots in the caption of Fig. 5 and improved text of the paper.

Sincerely yours,

Dr. N.V. Gavrilov on behalf of authors

Round 2

Reviewer 2 Report

The author has revised and supplemented the manuscript by reflecting the comments of the rviewer well.

Therefore, I judge that this manuscript can be published.